# Comparative analysis of COVID-19 vaccine responses and third booster dose-induced neutralizing antibodies against Delta and Omicron variants

Milja Belik [1✉], Pinja Jalkanen [1], Rickard Lundberg[1], Arttu Reinholm[1], Larissa Laine[2], Elina Väisänen[1,2], Marika Skön[2], Paula A. Tähtinen[3], Lauri Ivaska [3], Sari H. Pakkanen[4], Hanni K. Häkkinen[4], Eeva Ortamo[4], Arja Pasternack [5], Mikael A. Ritvos[5], Rauno A. Naves[5], Simo Miettinen[6], Tarja Sironen [6,7], Olli Vapalahti [6,7], Olli Ritvos[5], Pamela Österlund [2], Anu Kantele[4,9], Johanna Lempainen[1,3,8,9], Laura Kakkola [1,9], Pekka Kolehmainen [1,9] & Ilkka Julkunen [1,8,9✉]

Two COVID-19 mRNA (of BNT162b2, mRNA-1273) and two adenovirus vector vaccines (ChAdOx1 and Janssen) are licensed in Europe, but optimization of regime and dosing is still ongoing. Here we show in health care workers ($n = 328$) that two doses of BNT162b2, mRNA-1273, or a combination of ChAdOx1 adenovirus vector and mRNA vaccines administrated with a long 12-week dose interval induce equally high levels of anti-SARS-CoV-2 spike antibodies and neutralizing antibodies against D614 and Delta variant. By contrast, two doses of BNT162b2 with a short 3-week interval induce 2-3-fold lower titers of neutralizing antibodies than those from the 12-week interval, yet a third BNT162b2 or mRNA-1273 booster dose increases the antibody levels 4-fold compared to the levels after the second dose, as well as induces neutralizing antibody against Omicron BA.1 variant. Our data thus indicates that a third COVID-19 mRNA vaccine may induce cross-protective neutralizing antibodies against multiple variants.

[1] Institute of Biomedicine, University of Turku, Turku, Finland. [2] Finnish Institute for Health and Welfare, Helsinki, Finland. [3] Department of Paediatrics and Adolescent Medicine, Turku University Hospital and University of Turku, Turku, Finland. [4] Department of Infectious Diseases, Meilahti Vaccination Research Center, MeVac, Helsinki University Hospital and University of Helsinki, Helsinki, Finland. [5] Department of Physiology, University of Helsinki, Helsinki, Finland. [6] Department of Virology, University of Helsinki and HUSLAB, Helsinki, Finland. [7] Department of Veterinary Biosciences, University of Helsinki, Helsinki, Finland. [8] Clinical Microbiology, Turku University Hospital, Turku, Finland. [9]These authors contributed equally: Anu Kantele, Johanna Lempainen, Laura Kakkola, Pekka Kolehmainen, Ilkka Julkunen. ✉email: milja.j.belik@utu.fi; ilkka.julkunen@utu.fi

The emergence and global spread of severe acute respiratory syndrome coronavirus 2 (SARS-CoV-2) prompted a rapid development of vaccines to prevent SARS-CoV-2 infection and the coronavirus disease (COVID-19). Currently, the European Medicines Agency (EMA) has authorized four COVID-19 vaccines for use: mRNA-based BNT162b2 (Comirnaty, Pfizer-BioNTech) and mRNA-1273 (Spikevax, Moderna), and adenoviral vector-based ChAdOx1 (Vaxzevria, AstraZeneca) and Janssen COVID-19 Vaccine (Janssen). All four vaccines utilize SARS-CoV-2 spike protein as the antigen. In Finland, COVID-19 vaccinations started at the end of December 2020 with two doses of BNT162b2 administrated at a 3-week interval. Early 2021 vaccinations expanded to a larger population, and mRNA-1273 and ChAdOx1 vaccines were also included in the vaccination campaign. Also, the Finnish health care authorities made the decision to prolong the vaccination interval between the first and second doses from 3 weeks to 12 weeks. In March 2021, reports on rare cases of increased risk of blood clotting events associated with ChAdOx1[1,2] led to restrictions in the use of the vaccine, and many vaccinees with the first dose of ChAdOx1 received BNT162b2 or mRNA-1273 as the second dose. Recent studies have shown that heterologous 2-dose vaccinations with ChAdOx1 followed by BNT162b2 or mRNA-1273 elicit strong immune response[3,4] and higher levels of neutralizing antibodies than homologous 2-dose vaccinations with vector-based[5] or mRNA-based vaccines[6].

However, the antibody levels decline, and the surveillance of vaccine effectiveness and the decline of immune responses highlighted the need for a vaccine booster dose in the first group of vaccinees who received the vaccines within a 3-week interval[7,8]. Initial studies have shown that a third dose of COVID-19 vaccines strongly boost the waning immune responses and on the population level reduced severe COVID-19-associated morbidity and mortality[9–11]. Decisions for the third vaccine dose have been further promoted by the emergence of new SARS-CoV-2 variants-of-concerns (VOCs). Currently, the list of VOCs includes two widely circulating variants: Delta (B.1.617.2) and Omicron BA.1 (B.1.1.529). Delta variant has been globally dominant since spring 2021 replacing most of the other circulating SARS-CoV-2 variants[12,13], however, in November 2021, the first cases of Omicron BA.1 variant were reported, and only within one month Omicron BA.1 variant has spread globally initiating the replacement of the Delta variant[12,13]. In many countries the emergence of Omicron variants (BA.1 and BA.2) and previously observed waning humoral immunity has speeded up the recommendation for a third vaccine dose for all vaccinees.

Here we show, that two vaccine doses of four different vaccine combinations elicit SARS-CoV-2 spike-specific antibody responses with high, but subsequently declining, neutralizing titers against D614G and Delta variants. Importantly, the third vaccine dose significantly increases the waning antibody levels and the neutralization titers against the three variants, including the newly emerged Omicron BA.1 variant. Knowledge on the duration of vaccine-induced antibody responses by different vaccines and vaccine combinations is essential for making rational decisions regarding the timing, the number, and the combination of vaccine doses.

## Results

**Study population characteristics**. This study included a cohort of 328 health care workers (HCWs) who received two doses of COVID-19 vaccine either with a short 3-week interval (2.6-4.0 weeks) or a long 12-week interval (8.0–16.4 weeks; Table 1 and Fig. 1). All participants vaccinated with a short dose interval ($n = 120$) received two doses of BNT162b2 vaccine and a third booster vaccine dose of BNT162b2 ($n = 47$) or mRNA-1273 vaccine ($n = 73$). Participants vaccinated with a long dose interval received two doses of BNT162b2 ($n = 62$) or mRNA-1273 ($n = 72$) vaccines or a combination of ChAdOx1 and BNT162b2 ($n = 52$) or ChAdOx1 and mRNA-1273 ($n = 22$) vaccines. The cohorts vaccinated with the ChAdOx1 vaccine consisted of fewer participants since Finland restricted the use of ChAdOx1 vaccine in May 2021[1,2]. Although the five vaccine cohorts were different in size, the cohorts were demographically representative with each other (Table 1). Altogether, 87% were female (mean age 44 years) and 13% were male (mean age 46 years). Twelve participants had a PCR confirmed SARS-CoV-2 infection before the vaccination program.

Sequential serum samples were collected from all HCWs before vaccination and three weeks after receiving the first and the second vaccine doses (Fig. 1). From vaccinees with a short dose interval, the follow-up serum samples were collected three, six,

**Table 1 Demographics of COVID-19 vaccinated HCWs ($n = 328$).**

| | 2x BNT162b2 short dose interval | 2x BNT162b2 long dose interval | 2x mRNA-1273 long dose interval | ChAdOx + BNT162b2 long dose interval | ChAdOx1 + mRNA-1273 long dose interval |
|---|---|---|---|---|---|
| N | 120 | 62 | 72 | 52 | 22 |
| Female (%) | 100 (83.3%) | 54 (87.1%) | 67 (93.1%) | 44 (84.6%) | 21 (95.5%) |
| Male (%) | 20 (16.7%) | 8 (12.9%) | 5 (6.9%) | 8 (15.4%) | 1 (4.5%) |
| Previous PCR confirmed SARS-CoV-2 infections | 4 | 3 | 3 | 2 | 0 |
| **Age in years** | | | | | |
| Mean | 44 | 47 | 44 | 48 | 43 |
| Median | 44 | 50 | 43 | 50 | 43 |
| Range | 25–65 | 22–64 | 25–66 | 24–67 | 23–62 |
| **Mean time between vaccine doses (range)** | | | | | |
| Between 1st and 2nd dose in weeks | 3.0 (2.6–4.0) | 11.7 (8.0–15.0) | 12.1 (11.9–16.4) | 12.2 (10.9–16.0) | 12.2 (12.0–116.3) |
| Between 2nd and 3rd dose in months | 8.3 (7.6–9.3) | - | - | - | - |
| **Breakthrough infections after two vaccine doses** | | | | | |
| Based on PCR test | 1 | 0 | 0 | 0 | 0 |
| Based on antibody test | 0 | 0 | 0 | 1 | 0 |

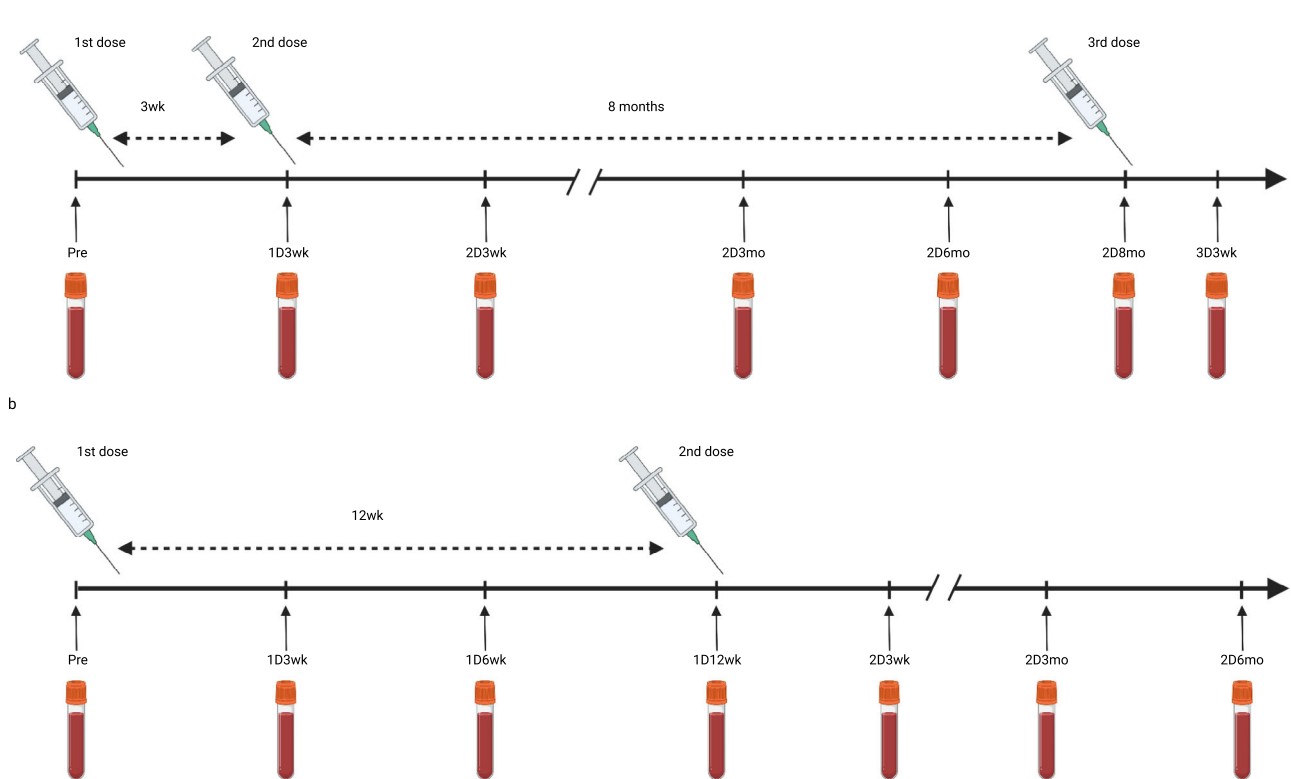

**Fig. 1 Timeline of vaccinations and samplings with 3-week and 12-week dosing intervals. a** In the short 3-week dose interval group sera were collected before vaccination (Pre), three weeks (1D3wk) after the first vaccine dose (1D), and three weeks (2D3wk), three months (2D3mo), six months (2D6mo), and eight months (2D8mo) after the second vaccine dose (2D), as well as 3 weeks (3D3wk) after the third vaccine dose (3D). **b** In the long 12-week interval dose group sera were collected before vaccination (Pre), three weeks (1D3wk), six weeks (1D6wk) and twelve weeks (1D12wk) after the first vaccine dose, and three weeks (2D3wk), three months (2D3mo) and six months (2D6mo) after the second vaccine dose. Created with BioRender.com.

and eight months after the second dose and three weeks after the third booster vaccine dose. From vaccinees who received at least one dose of BNT162b2 with a long dose interval, the follow-up serum samples were collected three and six months after the second vaccine dose, whereas from vaccinees in other groups only one follow-up serum sample was collected three months after the second vaccine dose (Supplementary Table 1).

**COVID-19 vaccine-induced antibody responses with a long vaccine dose interval**. To study the levels of antibodies elicited by four combinations of COVID-19 vaccines administered with a long 12-week vaccine dose interval, sequential serum samples were collected from HCWs vaccinated with a homologous combination of BNT162b2 or mRNA-1273, or with heterologous combination of ChAdOx1 as the first dose and either BNT162b2 or mRNA-1273 as the second dose. SARS-CoV-2 S1-specific IgG antibody levels were measured with enzyme immunoassay (EIA). Among the vaccinees who were seronegative before the vaccination, the production of anti-S1 IgG antibodies was induced at high level by BNT162b2, and mRNA-1273 vaccines compared with that of ChAdOx1 induced responses (Fig. 2). Three of the HCWs who were seronegative after the first ChAdOx1 vaccine dose were positive when tested with a lower serum dilution (Supplementary Fig. 1). Three weeks after the second dose, the antibody levels were significantly increased ($p < 0.0001$ or $p < 0.001$) in all vaccinee groups (Fig. 2). Furthermore, the age of the vaccinees (22–65 years) had little effect on the vaccine induced immune response after two doses (Supplementary Fig. 2). Three weeks after the second dose, geometric means were

128 EIA units for 2x BNT162b2, 158 for 2x mRNA-1273, 89 for ChAdOx1 + BNT162b2, and 153 for ChAdOx1 + mRNA-1273, indicating a high overall induction of antibody levels by the second immunization with all four vaccine combinations.

Vaccinees with long vaccine dose intervals and with prior SARS-CoV-2 infection ($n = 8$; diagnosed 25–393 days before the first serum sample, black dots in Fig. 2) developed high levels of S1-specific antibodies that exceeded the geometric mean value of all vaccinees after the first vaccine dose. In these vaccinees, the second dose maintained the S1-specific antibody levels or further increased them. Before vaccination, five of the vaccinees with prior SARS-CoV-2 infection had elevated nucleocapsid (N)-specific antibodies, as did five vaccinees with no diagnosed SARS-CoV-2 infection (Supplementary Fig. 3). N-specific antibodies are associated with a previous infection, and it is possible that the five participants with anti-N IgG antibodies but no PCR-confirmed SARS-CoV-2 infection had contracted SARS-CoV-2. The lack of N-specific antibody levels in three vaccinees with a prior PCR-confirmed SARS-CoV-2 infection may be explained by the long period between infection and the first sample collection (>301 days).

In all vaccinees, the antibody levels decreased gradually during the follow-up period. However, 3 months after the second vaccine dose 99% (170/172) of HCWs still had detectable levels of anti-S1 IgG antibodies (Fig. 2). Furthermore, two groups (2x BNT162b2 and ChAdOx1 + BNT162b2) were also analyzed 6 months after the second dose. In these vaccinees, although 94% (44/47) still had detectable levels of anti-S1 IgG antibodies, the antibody levels continued to decrease (Fig. 2). Regardless of this decrease, only one possible breakthrough infection was detected in the

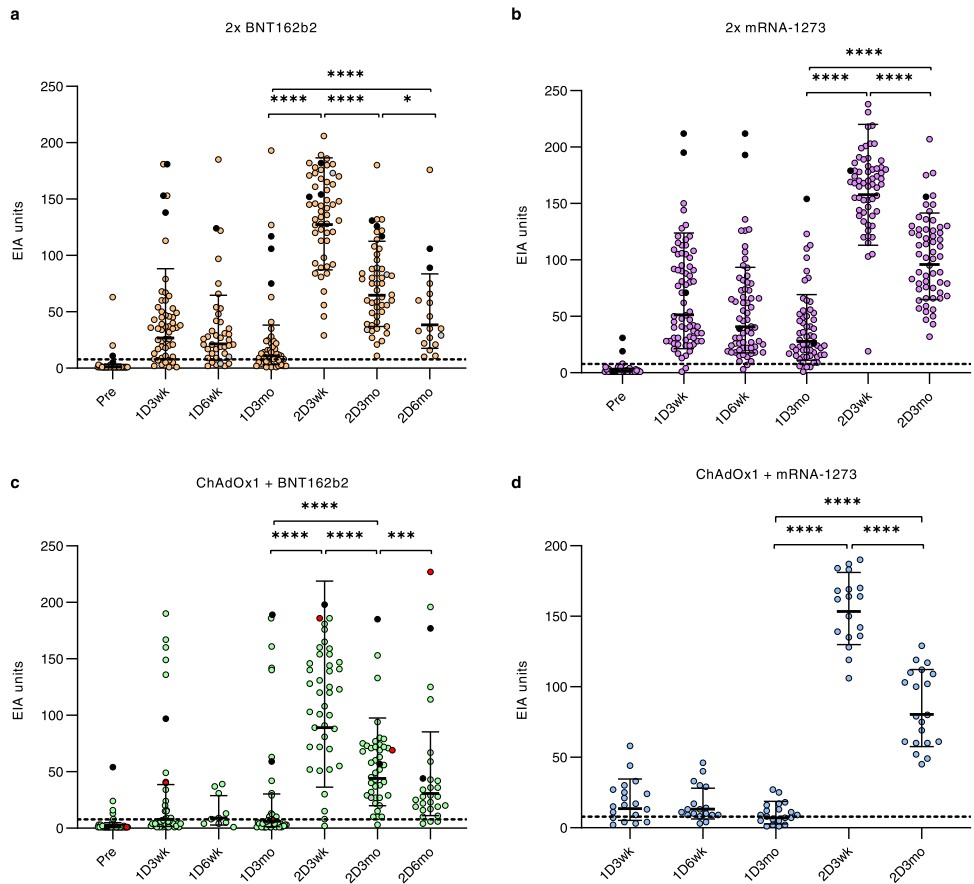

**Fig. 2 Antibody responses of HCWs after receiving one and two doses of COVID-19 vaccines with a long vaccine dose interval.** SARS-CoV-2 S1-specific IgG antibody levels were measured with EIA in all vaccine combination groups (**a** 2x BNT162b2, **b** 2x mRNA-1273, **c** ChAdOx1 + BNT162b2, **d** ChAdOx1 + mRNA-1273) from sera collected before vaccination (Pre), three weeks (1D3wk), six weeks (1D6wk) and twelve weeks (1D12wk) after the first vaccine dose (1D), and three weeks (2D3wk), three months (2D3mo), and six months (2D6mo) after the second vaccine dose (2D). The number of samples in each timepoint is presented in Supplementary Table 1. Eight HCWs with PCR confirmed SARS-CoV-2 infection prior to vaccinations are represented with black dots and one HCW with a breakthrough infection (>20 EIA unit increase between samples) after the second vaccine dose is represented with red dots. Geometric mean with geometric SD is represented in figures. Statistical differences between samples collected before vaccination vs. three weeks after the first vaccine dose, between samples collected three vs. twelve weeks after the first dose, between samples collected twelve weeks after the first dose vs. three weeks after the second dose, and between samples collected three weeks after both vaccine doses, were analyzed with two-sided Wilcoxon signed-rank test. P-values <0.05 were considered statistically significant. *P*-values: <0.0001 marked with ****, 0.0007 with *** and 0.0118 with *, ns indicates not significant. Cut-off values are indicated with dashed lines.

ChAdOx1 + BNT162b2 group as judged by the development of anti-N IgG antibodies after the second vaccine dose (red line in Supplementary Fig. 3 and Table 1). In addition, this vaccinee showed a high increase in S1-specific IgG levels between the 3- and 6-month samples collected after the second vaccine dose (red dots in Fig. 2). These results indicate high antibody responses induced by all vaccine combinations and a very low rate of breakthrough infections after two vaccine doses administered with a long vaccine dose interval during this follow-up period.

**BNT162b2 vaccine-induced antibody responses with a short vaccine dose interval and the effect of a third booster dose.** HCWs receiving 2x BNT162b2 vaccine with a short 3-week vaccine dose interval have been analyzed in our previous studies for the antibody levels at 6 weeks[14] (*n* = 180) and 6 months[7] (*n* = 52) after the second dose. Here we analyzed the antibody levels of 120 HCWs randomly selected from the above-mentioned cohort and extended the follow up to 9 months after the second vaccine dose. As shown also in our previous publications, the first vaccine dose induced the production of anti-SARS-CoV-2 S1 IgG antibodies, and the second dose significantly increased the

antibody levels (*p* < 0.0001) in naïve vaccinees (Fig. 3a). In contrast, vaccinees with a previous PCR-confirmed SARS-CoV-2 infection (*n* = 4, black dots and lines in Fig. 3a) mounted a high antibody response already after the first vaccine dose and the second vaccine dose further increased the antibody levels but only weakly. In all vaccinees, the antibody levels decreased gradually during the follow-up period and 7 to 9 months after the second vaccine dose 87% (95/105) of vaccinees had detectable levels of anti-S1 IgG antibodies. One vaccinee (0.8%) had a PCR-confirmed mild SARS-CoV-2 infection 47 days after the second vaccine dose (red dots and red line in Fig. 3a), however, this infection did not increase the S1-or N-specific antibody levels (red line in Supplementary Fig. 3). None of the initially naïve participants developed anti-N IgG antibodies after the second vaccine dose, while two of the vaccinees with a recent PCR-confirmed SARS-CoV-2 infection (diagnosed 16-30 days before the first serum sample) had N-specific antibodies (black dots in Supplementary Fig. 3). Thus, similar to the results of the long vaccine dose interval, the findings indicate a gradual decrease in S1-specific antibody levels and a very low rate of breakthrough infections after two vaccine doses with a short vaccine dose interval.

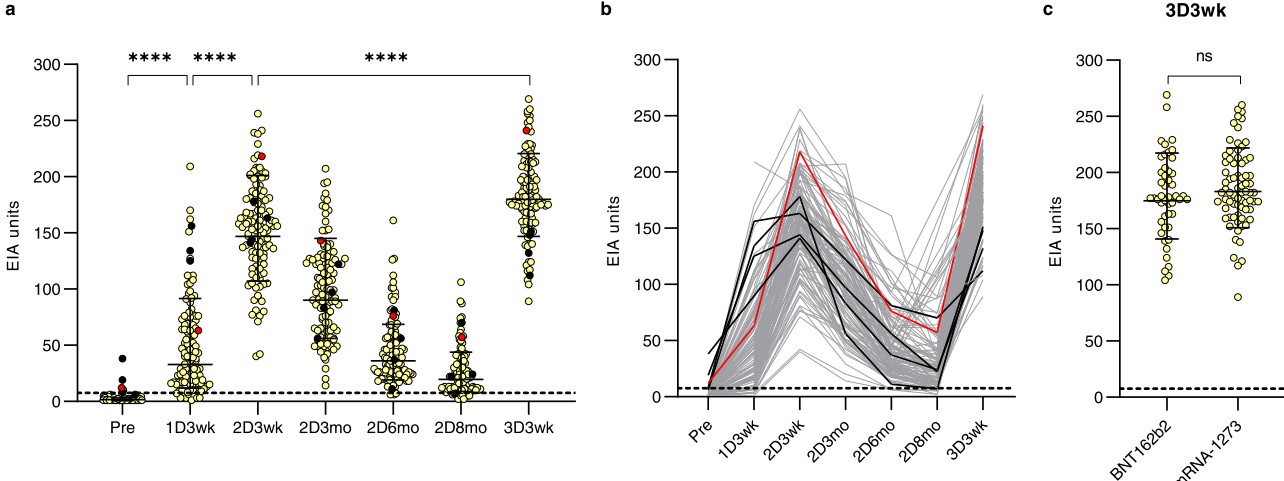

**Fig. 3 Antibody responses of HCWs after receiving two doses of COVID-19 mRNA vaccines with a short vaccine dose interval and the third booster dose of mRNA vaccines. a, b** SARS-CoV-2 S1-specific IgG antibody levels were measured with EIA from serum samples collected before the vaccination (Pre; $n = 108$), three weeks after the first vaccine dose (1D3wk; $n = 119$), three weeks (2D3wk; $n = 120$), three months (2D3mo; $n = 120$), six months (2D6mo; $n = 119$), and nine months (2D8mo; $n = 109$) after the second vaccine dose, and three weeks after the third booster vaccine dose (3D3wk, $n = 120$). Vaccinees received two doses of the BNT162b2 vaccine and the third dose of BNT162b2 or mRNA-1273. Sequential serum samples are connected with lines. HCWs with a prior PCR confirmed SARS-CoV-2 infection are represented with black lines and a PCR-confirmed breakthrough infection after the second dose is represented with a red line. The dotted line indicates the cut-off value for S1-based EIA. **c** Vaccinated HCWs are separated into two groups based on the third booster vaccine dose (BNT162b2, $n = 47$ vs. mRNA-1273, $n = 73$) and anti-S1 IgG antibody responses three weeks after the third booster dose is compared. Geometric mean antibody levels are shown as lines with geometric SD. Statistical differences between samples collected before vaccination, and three weeks after each vaccine dose were analyzed with two-sided Wilcoxon signed-rank test. Differences between different vaccine groups were compared with unpaired t-test. Two-tailed P-values < 0.05 were considered statistically significant. P-values of **** in panel a are <0.0001, ns not significant.

HCWs vaccinated with 2xBNT162b2 within a short 3-week dose interval ($n = 120$) received a third booster vaccine dose 8.3 months (7.6–9.3) after the second vaccine dose (47 received BNT162b2 and 73 received mRNA-1273 vaccine). Three weeks after the booster dose, the levels of SARS-CoV-2 S1-specific IgG antibodies were significantly higher than the levels seen after the second vaccine dose (geometric mean 147 and 180 EIA units three weeks after the second and the third dose, respectively) ($p < 0.0001$) (Fig. 3a). The booster dose induced slightly weaker immune response in older vaccinees (55–65 years) compared to 35–54-year-old vaccinees, and interestingly, lowest antibody levels were seen with the youngest vaccinees (20–34 years, $p < 0.0001$ compared to 34–44 years and 45–54 years; Supplementary Fig. 2). Both mRNA-based vaccines elicited similar booster effect since no difference was detected in antibody levels between the HCWs who received BNT162b2 or mRNA-1273 as a third dose (Fig. 3b).

**The effect of a third booster dose on neutralizing antibodies against Delta and Omicron variants.** In addition to anti-spike antibody levels, the capability of sera to neutralize SARS-CoV-2 variants was investigated. For this, we analyzed in-vitro the neutralization titers against currently circulating SARS-CoV-2 variants Delta and Omicron BA.1, and the ancestral D614G variant using isolated virus strains in microneutralization test (MNT). Figure 4c illustrates the amino acid changes between Delta and Omicron BA.1 variants.

Fifty-nine HCW vaccinees (no prior PCR-confirmed SARS-CoV-2 infection) who received two BNT162b2 vaccines within a short dose interval and a third booster dose were randomly selected from the study population. Neutralizing antibodies were analyzed in the serum samples collected at 3 weeks, 3 months, 6 months, and 8 months after the second vaccine dose and 3 weeks after the third booster dose (Fig. 4a).

Three weeks after the second vaccine dose, all vaccinees (100%, 59/59) neutralized the D614G, and 98% (58/59) and 24% (14/59) of vaccinees neutralized Delta and Omicron BA.1 variants, respectively. By 3 and 6 months after the second vaccine dose, the mean neutralization titers decreased by 2-fold against D614G and Delta, while the titers against Omicron BA.1 variant were virtually below the detection limit, except for the titers of the vaccinee who had a PCR-confirmed mild SARS-CoV-2 breakthrough infection (red dots in Fig. 4a). At the time of administration of the third booster dose (8 months after the second dose), 90% (53/59), 56% (33/59), and only 5% (3/59) of HCWs had neutralizing antibodies against D614G, Delta, and Omicron BA.1 variants, respectively. Strikingly, the booster dose induced strong responses against all three SARS-CoV-2 variants: three weeks after the third booster dose 100% of vaccinees' sera neutralized all three variants(Fig. 4b). The mean titers of neutralizing antibodies against D614G at 3 weeks after the second vaccine dose were 2.5- and 23-fold higher as compared to Delta, and Omicron BA.1, respectively. At 3 weeks after the third vaccine dose the differences between D614G vs. Delta, and Omicron BA.1 were 2- and 6-fold lower, respectively (Fig. 4b, Supplementary Fig. 4). Remarkably, regardless of the accumulating amino acid changes in the S1 of variants (Fig. 4c), sera of three-times mRNA-vaccinated HCWs showed strong neutralizing capability against all three variants.

**Comparison of COVID-19 vaccine-induced antibody responses after 2-dose vaccination with different vaccine combinations.** To analyze differences in antibody levels elicited by five different vaccine combinations and by different vaccine dose intervals, spike-specific antibody levels in sera from HCWs vaccinated with 2x BNT162b2 with short and long vaccine dose intervals ($n = 120$ and $n = 53$, respectively), 2x mRNA-1273 ($n = 60$), ChAdOx1 + BNT162b2 ($n = 43$) and ChAdOx1 + mRNA-1273 ($n = 20$),

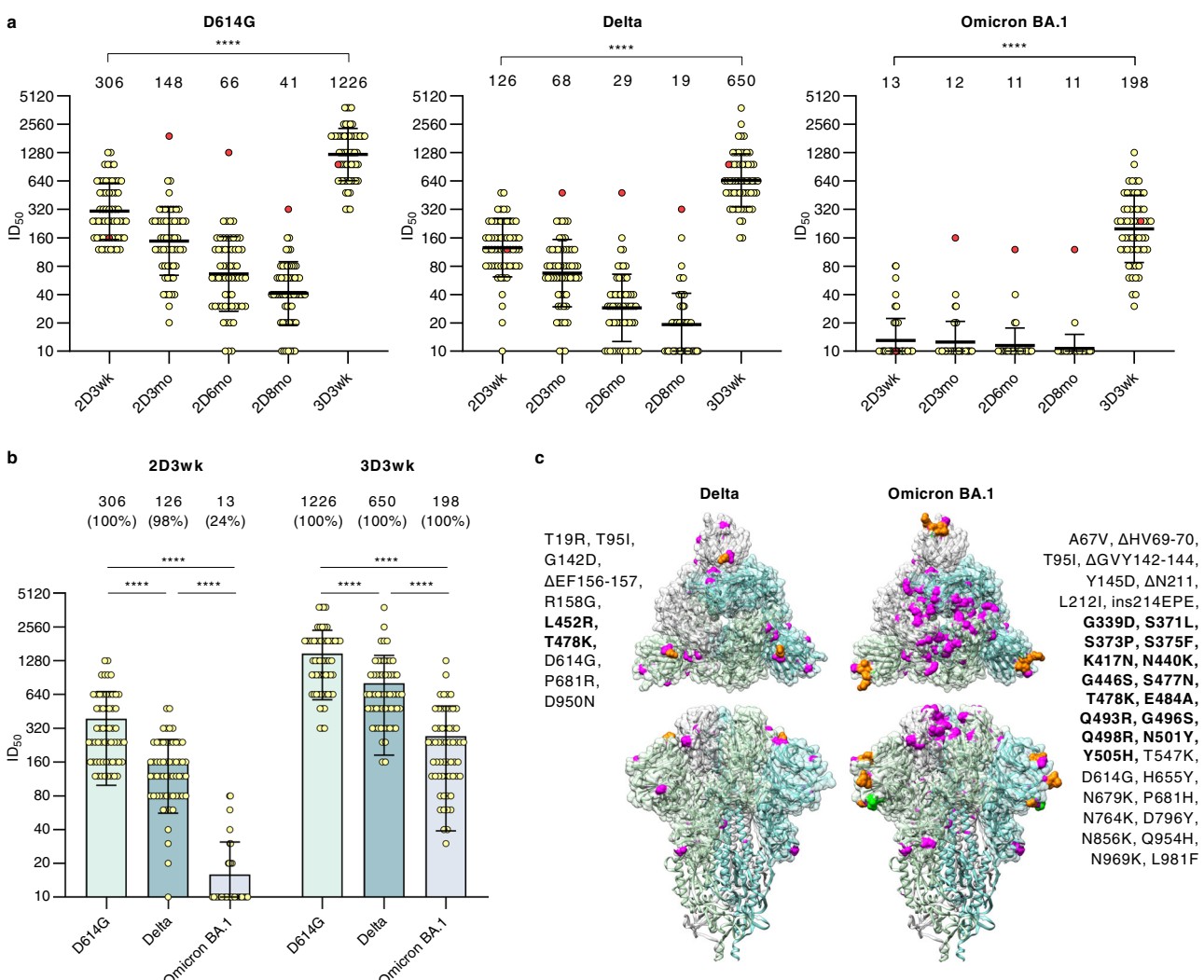

**Fig. 4 Neutralizing antibodies of HCWs receiving three doses of BNT162b2 vaccine. a** Neutralizing antibody responses of HCWs ($n = 59$) were analyzed with a microneutralization test against SARS-CoV-2 isolates representing D614G, Delta and Omicron BA.1 variants. Serum samples were collected before the first vaccine dose and three weeks after the second and the third vaccine dose. Additionally, follow-up serum samples were collected 3, 6, and 8 months after the second vaccine dose. A vaccinee with a PCR-confirmed SARS-CoV-2 infection after vaccination is marked with red dots. **b** Neutralization titers against D614G, Delta, and Omicron BA.1 variants after the second and third vaccine doses were compared. Geometric mean, shown as a line with geometric SD, is indicated above the figure with the percentage of positive samples when neutralization titer ≥20 is considered positive. Differences between neutralization titers were analyzed with two-sided Wilcoxon signed-rank test and p-values <0.05 were considered statistically significant. *P*-values <0.0001 are marked with ****. **c** Amino acid changes in Delta and Omicron BA.1 compared to Wuhan Hu-1 sequence are shown in trimeric SARS-CoV-2 spike protein structure (PDB: 6VXX). Substitutions are highlighted with magenta and deletions with orange. Changes located in the receptor-binding domain are bolded.

collected 3 weeks and 3 months after the second doses were compared (Fig. 5a). In all vaccine combination groups, the antibody levels were elevated 3 weeks after vaccination and decreased by 3 months after the vaccination. The rate of decline from 3 weeks to 3 months was similar between all vaccine combinations (fold-decrease 1.6-2.0). The vaccine groups including mRNA-1273 as the second dose presented the highest geometric mean antibody levels 3 weeks and 3 months after vaccination (geometric means 153–158 and 80–96, respectively), while the vaccine groups including the BNT162b2 vaccine showed the lowest geometric mean antibody levels after vaccination (geometric means 89-147 and 44-90, respectively).

A representative number of serum samples from the vaccinees in five vaccine combination groups (2x BNT162b2 with short dose interval, $n = 59$; 2x BNT162b2 with long vaccine dose interval, $n = 29$; 2x mRNA-1273, $n = 30$; ChAdOx1 +

BNT162b2, $n = $; ChAdOx1 + mRNA-1273, $n = 20$) was analyzed at 3 weeks and 3 months after the second vaccination for the neutralization capacity against D614G and Delta variants (Fig. 5b, c). The vaccine group of 2x BNT162b2 with a short vaccine dose interval showed a 2-3x lower GMT value against the D614G three weeks after second vaccine dose compared to the groups vaccinated with a long vaccine dose interval: 2x BNT162b2 long ($p < 0.0001$), 2x mRNA-1273 ($p < 0.0001$), ChAdOx1 + mRNA-1273 ($p < 0.0001$) and ChAdOx1 + BNT162b2 ($p < 0.0014$). Three months after vaccination, the differences in GMTs between groups against D614G diminished, and only the group with 2x mRNA-1273 vaccine had a higher GMT than the group vaccinated with 2x BNT162b2 with a short vaccine dose interval ($p < 0.0001$) (Fig. 5b).

Neutralization titers against the Delta variant was the highest in the group vaccinated with 2x mRNA-1273 (GMT 339 at three

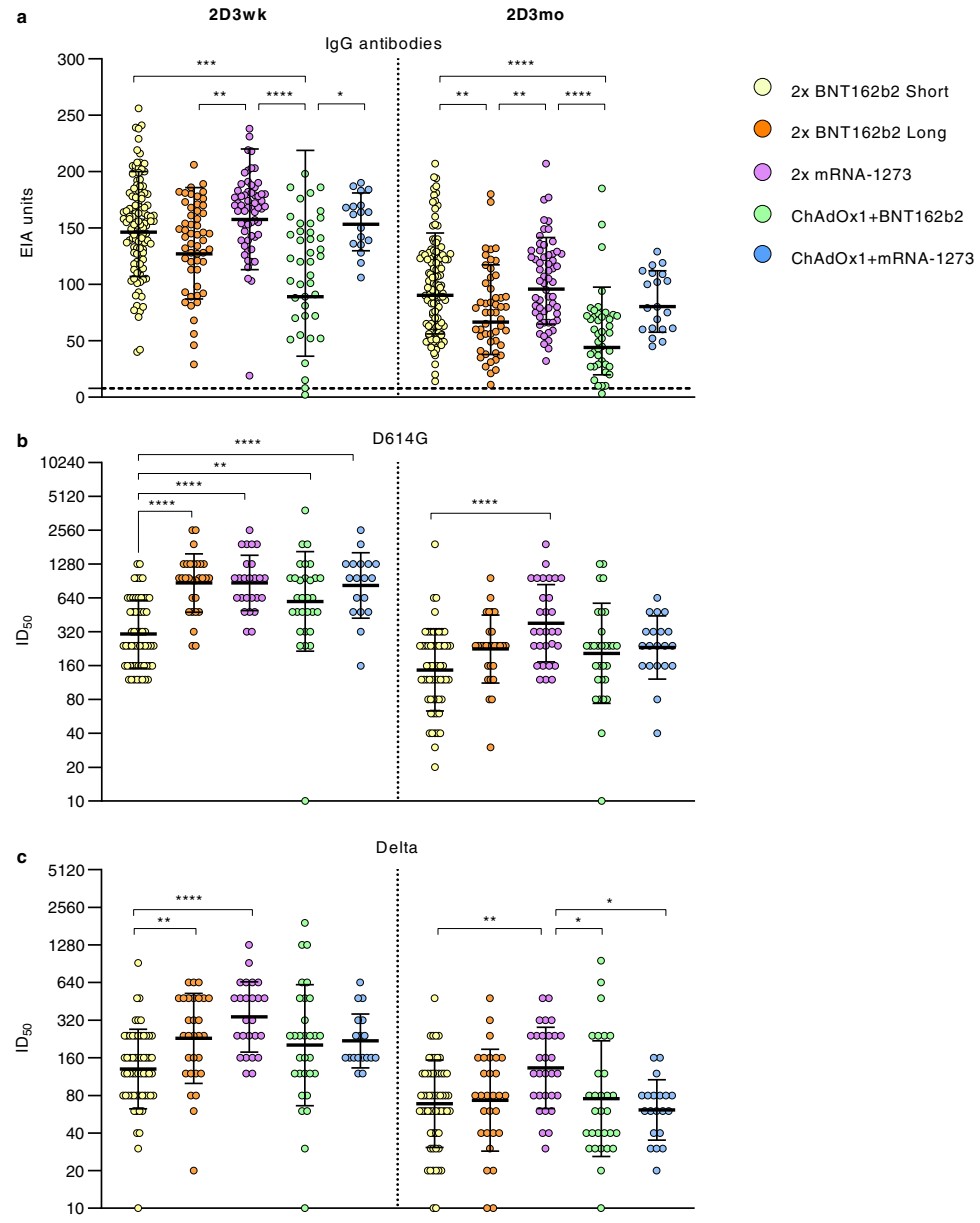

**Fig. 5 SARS-CoV-2 neutralizing antibodies after two doses of different combinations of COVID-19 vaccines. a** Serum samples collected 3 weeks (2D3wk) and three months (2D3mo) after the second vaccine dose from HCWs who were vaccinated with 2x BNT162b2 with a short (n = 120) or a long vaccine dose interval (n = 59), 2x mRNA-1273 (n = 67), ChAdOx1 + BNT162b2 (n = 47), or ChAdOx1+mRNA-1273 (n = 21) were compared for anti-SARS-CoV-2 S1 IgG antibodies (EIA units). The dotted line indicates the cut-off value. Neutralizing antibody responses against **b** D614G and **c** Delta variants were analysed for vaccinees with two doses of vaccine combinations: 2x BNT162b2 with a short (n = 59), or a long vaccine dose interval (n = 29), 2x mRNA-1273 (n = 25 at 2D3wk and 30 at 2D3mo), ChAdOx1 + BNT162b2 (n = 30), or ChAdOx1+mRNA-1273 (n = 18 at 2D3wk and 20 at 2D3mo). Half-maximal inhibitory dilutions (ID50) were calculated and titers <20 were marked as 10. Geometric mean titers (GMTs) for each vaccine group are shown as lines with geometric SDs. Differences in IgG antibody levels and neutralization titers between different groups were analysed with two-sided Kruskal–Wallis rank test and P-values <0.05 were considered statistically significant. P-values: <0.0001 are marked with ****, from right to left in panel **a** 0.0001 ***, 0.0019 **, and 0.0215 *, from right to left in panel **b** 0.0023 ** and ** 0.0014, and from right to left in panel **c** 0.0096 **, 0.0098 **, 0.0307 *, and 0.0122 *.

weeks and GMT 133 at three months after vaccination) (Fig. 5). At three weeks after vaccination, the GMTs in two vaccine groups with a long vaccine dose interval, 2x mRNA-1273 and 2x BNT162b2 (long), were significantly higher than in the 2x BNT162b2 short dose interval group (p < 0.0001 and p < 0.0096). However, by three months after vaccination the GMTs against Delta variant decreased to relatively similar levels in all groups. Only the GMT of 2x mRNA-1273 group remained slightly higher than 2x BNT162b2 short (p < 0.0098), ChAdOx1 + BNT162b2 (p < 0.0307), and ChAdOx1 + mRNA-1273 (p < 0.0122). Three

months after the second vaccination, vaccinees without detectable neutralizing antibodies against Delta variant were detected in all vaccine groups which obtained BNT162b2 as the second dose (n = 1 in ChAdOx1 + BNT162b2, n = 2 in 2x BNT162b2 long vaccine interval, and n = 2 in 2x BNT162b2 short vaccine interval). One vaccinee (in group ChAdOx1 + BNT162b2) showed no neutralizing activity against D614G or Delta variants three weeks and three months after vaccination. Altogether, our results show that the studied vaccine combinations elicit high levels of SARS-CoV-2 antibodies.

## Discussion

SARS-CoV-2 has spread throughout the world and caused millions of deaths, and tremendous social and economic damage. The fast development of efficient COVID-19 vaccines has been an important measure against the pandemic the past year, and data is accumulating on how to utilize the licenced vaccines in the best possible way. New variants escaping infection or vaccine-induced immunity are of great concern, and this year the world has witnessed the appearance of Delta variant followed by the currently spreading Omicron BA.1 and BA.2 variants, which show a strong transmission ability leading to an upsurge of SARS-CoV-2 infections in numerous countries.

Initially, the vaccinations were administered with a 3-week dose interval, and later this was prolonged to several weeks (up to 12 weeks) depending on the country. In addition, due to a varying availability of vaccines, different combinations of vaccine doses have been used. It has been a matter of great concern and debate, how the timing and the various combinations of vaccines affect the immune responses. We show here that the five combinations of vaccines used in Finland resulted in similar antibody responses, with the mRNA-1273 vaccine eliciting slightly higher antibody responses. Similar results have been obtained by others[3–6], indicating that both heterologous and homologous vaccinations elicit proper immune response in healthy individuals.

Interestingly, our data shows somewhat higher neutralizing antibody levels in vaccinees who received mRNA-1273 in comparison to those who received BNT162b2 or ChAdOx1. This result is in line with the vaccine effectiveness studies where mRNA-1273 has been shown to be somewhat more efficient than BNT162b2 in preventing COVID-19[15,16]. This is well in line with the concept that neutralizing antibodies against SARS-CoV-2 correlate with a protection against symptomatic SARS-CoV-2 infection[17]. As for the timing of dosages, previous studies have shown that vaccine combinations given with long dosing intervals elicit higher or similar antibody responses compared to BNT162b2 with a short dosing interval[15,16]. The longer vaccine dosing interval has also resulted in at least equally high vaccine effectiveness as that seen with a short dosing interval of BNT162b2[15,16]. Similarly, according to our results, sera collected three months after the second vaccination with 3-week and 12-week dosing intervals, shows comparable neutralization capacity. Thus, it is likely that any COVID-19 vaccine can be combined for an effective vaccination, enabling efficient use of existing vaccine stores. This could also facilitate the development of new vaccine candidates, since it is likely that combinations of any highly immunogenic COVID-19 vaccines can be utilized for effective protection against a severe disease. We would like to point out that our study is an unbiased analysis of licenced vaccines, and the study has not received funding from the vaccine manufacturers.

The currently spreading Omicron BA.1 and BA.2 variants contain more than 30 amino acid changes in the spike protein, many of which have been associated with increased infectivity and antibody evasion in previous variants of concern[18]. Preliminary studies implicate, that antibodies elicited by infection with the original SARS-CoV-2 variant[19] or by two vaccine doses, have markedly reduced capability to neutralize Omicron (BA.1) variant[20]. Our data clearly confirms this initial observation that two doses of BNT162b2 vaccine does not provide good long-term immunity against the Omicron BA.1 variant. However, the third vaccine dose of BNT162b2 or mRNA-1273 induced strong immune responses against the ancestral D614G variant and high levels of cross-neutralizing antibodies against the Delta and Omicron BA.1 variants. This observation supports the concept that the present vaccines, as long as they are administered in sufficient doses (at least 3 doses), induce adequate protective immunity against the Omicron variants. It is also likely that cell-mediated immunity induced by vaccines contributes to protective efficacy against all VOCs. In subsequent analysis it will be of great interest to measure cell-mediated immunity against the emerging VOCs.

It has been suggested that ca. 10–20 substitutions in the spike protein are sufficient to create a variant that escapes neutralizing antibodies elicited by vaccinations. However, individuals with a previous SARS-CoV-2 infection followed by a subsequent vaccination efficiently neutralize these highly mutated spike proteins[21]. It has been concluded that the breadth and the amount of antibodies produced by memory B-cells is higher when anti-SARS-CoV-2 immunity is based on previous infection combined with a vaccination as compared to vaccination or infection alone[22,23]. Vaccinations alone seem to result in high levels of antibodies with a more limited breadth in neutralizing capacity, indicating the need for booster vaccinations to induce high antibody levels for providing maximal protective efficacy against the infection. Our data is well in line with this, showing that the third vaccine dose significantly increases the antibody levels and the neutralizing capacity, particularly against Omicron BA.1 variant, strongly indicating the benefits of a third vaccine dose in containment of the SARS-CoV-2 pandemic.

## Methods

**Study population.** Health care workers (HCWs) who completed a full two-dose vaccine regimen with a long 12-week (8.0-16.4 weeks) dose interval ($n = 208$) were included in the study. HCWs who completed a full three-dose vaccine regimen with a short 3-week (2.6–4.0 weeks) dose interval between the first two doses ($n = 120$) were selected from a larger cohort[14]. Vaccinees with three vaccine doses were followed up to 3 weeks after the third dose, and vaccinees in the long vaccine dose group were followed for a maximum of six months after the second vaccine dose. Serum samples were collected at regular time points (Supplementary Table 1). Participants filled symptom questionnaires before every sample collection and, if symptomatic, were encouraged to take a COVID-19 RT-qPCR test which was arranged as part of local infection control practice. The demographics of the vaccinated HCWs are presented in Table 1. The timeline of vaccinations and sample collection is presented in Fig. 1.

**SARS-CoV-2 S1- and N-protein based immunoassays.** SARS-CoV-2 spike subunit 1 (S1) and nucleocapsid (N) protein-specific antibodies were analyzed with an in-house enzyme immunoassay (EIA)[24]. Briefly, purified recombinant SARS-CoV-2 antigens were coated on 96-well plates (2.0 μg/ml of N and 3.5 μg/ml of S1). IgG levels of serum samples (diluted 1:1000) were determined with absorbance measurement at 450 nm wavelength. EIA results were also confirmed with a 1:300 serum dilution (Supplementary Fig. 1 and 3). Optical density (OD) values were converted to EIA units using linear interpolation between OD-values of a positive (=100 EIA units) and a negative control (=0 EIA units) serum specimen. Thresholds to determine seropositivity were calculated as described previously[14].

**SARS-CoV-2 variants.** SARS-CoV-2 isolates FIN25-20 (B.1, D614G variant, MW717675.1 and EPI_ISL_412971)[25], FIN37-21 (B.1.617.2, Delta variant, MZ945494 and EPI_ISL_2557176)[26] and FIN55-21 (B.1.1.529, Omicron BA.1 variant, EPI_ISL_8768822.2) were isolated from SARS-CoV-2 PCR-positive nasopharyngeal samples by incubation with VeroE6 (for FIN25-20) or VeroE6-TMPRSS2-H10 cells[27] (for FIN37-21 and FIN55-21) and further passaged in VeroE6-TMPRSS2-H10 cells in DMEM supplemented with 2% FBS, 2 mM L-glutamine, and penicillin-streptomycin. Titration of virus stocks was done using Median Tissue Culture Infectious Dose ($TCID_{50}$) assay.

**Microneutralization test.** The neutralization capacity of the serum samples was measured by microneutralization test (MNT) as described previously[7]. Briefly, two-fold dilution series starting from 1:10 dilution was prepared on 96-well plate for each serum into 50 μl of DMEM supplemented with 2% fetal bovine serum (FBS), 2 mM L-glutamine, and penicillin-streptomycin. Serum dilutions were incubated with 50 $TCID_{50}$ of virus in a total volume of 100 μl for 1 h at +37 °C (final dilution 1:20) before the addition of 50 000 VeroE6-TMPRSS2-H10 cells into the virus-serum dilution mixture to a final volume of 150 μl. The cells were incubated at +37 °C, 5% $CO_2$, for 4 days, fixed with 4% formaldehyde, stained with crystal violet and visualized for cell death. Reciprocal of serum dilution inhibiting 50% of cell death was determined as the neutralization titer. A serum was considered positive for neutralizing antibodies if it inhibited 50% of cell death at a dilution of 1:20 or above. Serum with known neutralizing titer was used as a control on each plate.

**Ethical statement**. Study participants were recruited among health care personnel of Turku University Hospital (TUH, Turku, Finland) (Southwest Finland health district ethical permission ETMK 19/1801/2020, EudraCT 2021-004419-14) and Helsinki University Hospital (HUH, Helsinki, Finland) (Helsinki-Uusimaa health district ethical permission HUS/1238/2020, EudraCT 2021-004016-26) prior to receiving COVID-19 vaccines as part of hospital occupational health care. At enrollment, written informed consent was collected from all participants.

**Statistical analysis, and illustration of changes in spike structure**. Data was collected in Excel 2016 (Microsoft 365) and analyzed in Prism v8 (GraphPad Software). Paired samples were tested with Wilcoxon signed-rank test. Differences between vaccine groups were tested with unpaired t-test or Kruskal–Wallis test followed with Dunn's multiple comparisons test. All tests were two-sided and p-values <0.05 were considered statistically significant. Changes in SARS-CoV-2 spike structure (PDB: 6VXX) were illustrated with UCSF Chimera v1.15 (RBVI, University of California).

**Reporting summary**. Further information on research design is available in the Nature Research Reporting Summary linked to this article.

## Data availability
All data are available upon request from the corresponding authors. Source data are provided with this paper. The SARS-CoV-2 sequences and protein data are available under accession codes PDB ID 6VXX, GenBank IDs MW717675.1 [https://www.ncbi.nlm.nih.gov/nuccore/MW717675] and MZ945494, and GISAID IDs EPI_ISL_412971 [https://www.epicov.org/epi3/frontend#81332], EPI_ISL_2557176 [https://www.epicov.org/epi3/frontend#39b08a], and EPI_ISL_8768822.2 [https://www.epicov.org/epi3/frontend#5d6de7].

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

## Acknowledgements
We thank Soili Jussila, Anne Suominen, Anne-Mari Pieniniemi, and Outi Debnam for technical assistance. This study was supported by the Academy of Finland (grant number 337530 to I.J., 336439 and 335527 to A.K., and 339512 to L.K.), Jane and Aatos Erkko Foundation (grant numbers 3067-84b53 and 5360-cc2fc to I.J., and 5359-89ef7 to O.V.), the Finnish Medical Foundation (to A.K.), the Sigrid Jusélius Foundation (to I.J. and L.K.), Juho Vainio Foundation (to O.V.), and HUH Funds (TYH2021343 to O.V.).

## Author contributions
M.B., P.J., P.K., L.K., J.L., A.K., and I.J. designed the experiments; M.B., P.J., P.K., R.L., A.R., and L.K. did microneutralization tests and analyzed the data; M.B., P.J., and P.K. did EIA tests and analyzed the data; L.L., E.V., M.S., and P.Ö. isolated and characterized the virus isolates; H.K.H., S.H.P., P.A.T., E.O., S.M., T.S., O.V., L.I., J.L. and A.K. recruited vaccinees and patients and collected their sera and data; A.P., M.A.R., R.A.N., P.J. and O.R. produced antigens for EIA; M.B., P.J. and P.K. analyzed all data sets; M.B., P.J., P.K., L.K., and I.J. wrote the manuscript and all co-authors contributed to the edition of the text.

## Competing interests
The authors declare no competing interests.
