## [Peer Review File · Nature Communications]

REVIEWERS' COMMENTS

Reviewer #1 (Remarks to the Author):

In this manuscript, Belik et al characterize the humoral immune response after receiving COVID-19 vaccines. The authors collected and analyzed longitudinal serum samples from health care workers who received the BNT126b2, mRNA-1273, or the ChAdOx1 vaccines. Different combinations of vaccines along with different vaccine schedules were compared. The authors found that two doses of any of the vaccines elicited robust antiviral antibody titers. Antibody titers appear to decline at a similar rate regardless of the vaccine (although further analysis could be included to strengthen this point). Surprisingly, neutralizing antibody titers were very low against the circulating omicron VOC after two doses of the vaccines, and these serum neutralizing titers were significantly elevated after a third dose. These findings will be of broad interest and will influence the recommended vaccines and schedules into the future.

Overall this study is very well written and the data strongly support the authors conclusions. I only have the following minor critique.

Can the authors model the antibody titer rate of decay across the different vaccines using the sequential serum samples? The authors could focus on the decay rate after the second vaccine dose. This data could be useful to compare immune durability across the different vaccine constructs.

Reviewer #2 (Remarks to the Author):

Dear Authors,

I have read with interest the study analyzing comparative humoral immune results of different COVID19 vaccines. The study is well planned in a basis of longitudinal follow-up. Although there are many challenging factors for the presentation of the results, the results has been given considerably in a clear way.

Below you can find my suggestions regarding the manuscript:

1. The title is long, albeit sufficient to give the subject.
2. Abstract gives the required data in a precise way.
3. Introduction:

There are erroneously written 'and's in the first paragraph.

Line 47: Vaccination interval was prolonged from depending on what? Previous clinical trials? Reference should be given.

Line52: immune responses-> immune response
higher and similar levels ... than... Sentence should be corrected.

Line 57: who received the vaccine with.. -> vaccines within...

Line 59: reduce -> reduced

4. Results:

The titles of the Results may be more standard. Some are conclusive sentences whereas others not.

Line 91-92: The reason for restricted use of ChAdOx1 vaccine may be given by giving reference.

Line 109: ...in higher levels... ->at high level..

Line 115: 2D3wk -> 2days 3 weeks. Why did the authors write this in the parenthesis together with Fig 2.

Line 117: Possibly due to the transfer of the Material and Methods to the last part, some of the abbreviations, such as EIA and GMT, are not defined at first sight.

Line 117: 127... 158... The units are missing.

Line 126-128: Low N (nucleocapsid(please add!))-specific antibody levels... Why do the authors say 'low' here? Please define the 'low N-specific antibody levels'. Nucleocapsid-specific antibody levels associates with previous infection as the vaccine induces only S-specific antibody levels.

Do you know the anti-N specific antibody levels in vaccinees, especially those experienced previous COVID19 infection before vaccination and those having breakthrough infection? If so, these data may be given.

Line 155-156: Did the authors analyzed this vaccinee for a possible immunodeficiency?

Line 164: ...7.6-9.3 months... Please give the mean or median here.

Line 179: A new paragraph is needed for the second sentence beginning with 'Fifty-nine HCW...'

Line 219: Please explain GMT. I could not find the abbreviation explanation also in Figure 5. Please refer to the Figure here and at the end of this paragraph (Line 225).

5. Discussion:

Line 243: containment of the pandemic-> this phrase may be better changed.

6. Materials and Methods

Please define the N and S1-specific antibody levels.

What does Spike '1' stands for? Why did you add 1 at the end?

Line 321:as described earlier.-> 'before or previously' may be better than earlier.

Line 329: Median Tissue Culture Infectious Dose (TCID50) assay

Line 339: Reciprocal of serum dilution?

Line 355: UCSF Chimera 1.15 (program....date or version)

Reviewer #3 (Remarks to the Author):

The study "Comparative analysis of BNT162b2, mRNA-1273 and ChAdOx1 COVID-19 vaccine induced antibody responses and the 3rd BNT162b2 vaccine induced neutralizing antibodies against Delta and Omicron variants" by Belik et al is a descriptive study on antibody-responses to Covid-19 vaccination. The authors assess binding- and neutralizing antibody responses to SARS-CoV-2, in vitro, in serum from a cohort of HCWs after covid-19 vaccination.

Major concern:

It has already been demonstrated that a 3rd dose is beneficial for in vitro neutralizing antibodies against SARS-CoV-2 VoCs, including the Omicron variant (for example Pajon et al NEJM 2022, Garcia-Beltran, Cell 2022, etc). The manuscript by Belik et al verifies these results but do not provide additional in depth information of vaccine-induced immune responses to VoCs.

To understand if the differential dosing intervals matter for epitope specificity after vaccination, it would have been interesting to see a detailed dissection of S-specific B- and T cell responses.

The authors should acknowledge the discrepancy between protection from infection and protection from severe/fatal covid-19. I.e. that all vaccines appear quite efficient to protect against severe or

fatal covid-19 and that this protection is not correlating to antibody levels (whereas protection from infection does).

REVIEWERS' COMMENTS

We thank the Editor and the Reviewers for the careful evaluation of the manuscript and for the excellent comments to further improve our manuscript. We have now replied point-by-point to all the comments. We hope this revised version would be now suitable for publication in Nature Communications.

Reviewer #1 (Remarks to the Author):

In this manuscript, Belik et al characterize the humoral immune response after receiving COVID-19 vaccines. The authors collected and analyzed longitudinal serum samples from health care workers who received the BNT126b2, mRNA-1273, or the ChAdOx1 vaccines. Different combinations of vaccines along with different vaccine schedules were compared. The authors found that two doses of any of the vaccines elicited robust antiviral antibody titers. Antibody titers appear to decline at a similar rate regardless of the vaccine (although further analysis could be included to strengthen this point). Surprisingly, neutralizing antibody titers were very low against the circulating omicron VOC after two doses of the vaccines, and these serum neutralizing titers were significantly elevated after a third dose. These findings will be of broad interest and will influence the recommended vaccines and schedules into the future.

Overall this study is very well written and the data strongly support the authors conclusions. I only have the following minor critique.

Can the authors model the antibody titer rate of decay across the different vaccines using the sequential serum samples? The authors could focus on the decay rate after the second vaccine dose. This data could be useful to compare immune durability across the different vaccine constructs.

We thank Reviewer 1 for the encouraging comments. The Reviewer highlights the attractiveness of decay rate calculations that would allow comparison of the persistence of antibodies. We agree that modeling of the decay rate would be interesting and could possibly provide valuable information. We do, however, feel that the time series in the present study is too limited for such analysis since only one sample was collected at the decay phase from vaccinees receiving homologous or heterologous mRNA-1273 vaccination. Especially since the antibody decay can start fast and slow down later, the resulting decay rates would not give a reliable estimate of the shape of the antibody decay curve. We hope to answer this question of decay rates in forthcoming studies with the more comprehensive longitudinal follow-up where samples are collected for several months after vaccinations.

Reviewer #2 (Remarks to the Author):

Dear Authors,

I have read with interest the study analyzing comparative humoral immune results of different COVID19 vaccines. The study is well planned in a basis of longitudinal follow-up. Although there are many challenging factors for the presentation of the results, the results has been given considerably in a clear way.

We thank the Reviewer 2 for the encouraging and excellent comments.

Below you can find my suggestions regarding the manuscript:

1. The title is long, albeit sufficient to give the subject.

We have now modified the title as suggested by the Editor and the new title is "Comparative analysis of COVID-19 vaccine responses and third booster dose-induced neutralizing antibodies against Delta and Omicron variants".

2. Abstract gives the required data in a precise way.

We have now modified the abstract as suggested by the Editor and we hope the abstract presents the data now in a required detail.

3. Introduction:

There are erroneously written 'and's in the first paragraph.

We apologize for these typing errors and we have now corrected the errors.

Line 47: Vaccination interval was prolonged from depending on what? Previous clinical trials? Reference should be given.

This decision was made by the Finnish Ministry of Social Affairs and Health based on the recommendations by the group of vaccine experts. We have now corrected this sentence to "Also, the Finnish health care authorities made the decision to prolong the vaccination interval between the first and second doses from 3 weeks to 12 weeks."

Line 52: immune responses-> immune response higher and similar levels ... than... Sentence should be corrected.

Line 57: who received the vaccine with.. -> vaccines within...

Line 59: reduce -> reduced

We apologize for these typing errors and we have now corrected the errors.

4. Results:

The titles of the Results may be more standard. Some are conclusive sentences whereas others not.

We have now modified the titles to be of same style.

Line 91-92: The reason for restricted use of ChAdOx1 vaccine may be given by giving reference.

We have now added references 1 and 2 to indicate the rationale behind the decision made by the Finnish health authorities.

Line 109: ...in higher levels... ->at high level..

Line 115: 2D3wk -> 2days 3 weeks. Why did the authors write this in the parenthesis together with Fig 2.

Line 117: Possibly due to the transfer of the Material and Methods to the last part, some of the abbreviations, such as EIA and GMT, are not defined at first sight.

We apologize the typing errors and the incorrect placement of the definitions of abbreviations. We have now modified the text accordingly and removed 1D3wk, 2D3wk, 2D3mo, and 2D6mo when referring to Figure 2 in the text. In addition, we have defined EIA and GMT in the results section.

Line 117: 127... 158... The units are missing.

We have now added the missing units to all these sentences.

Line 126-128: Low N (nucleocapsid(please add!))-specific antibody levels... Why do the authors say 'low' here? Please define the 'low N-specific antibody levels'. Nucleocapsid-specific antibody levels associates with previous infection as the vaccine induces only S-specific antibody levels.

We have now added "nucleocapsid" on line 118 where N-specific antibodies are mentioned for the first time. In addition, we have now clarified the different anti-N antibody results on lines 119-123 "N-specific antibodies are associated with previous infection, and it is possible that the five participants with anti-N IgG antibodies but no PCR-confirmed SARS-CoV-2 infection had contracted SARS-CoV-2. The lack of N-specific antibody levels in three vaccinees with prior PCR-confirmed SARS-CoV-2 infection may be explained by the long period between infection and the first sample collection (>301 days).".

Do you know the anti-N specific antibody levels in vaccinees, especially those experienced previous COVID19 infection before vaccination and those having breakthrough infection? If so, these data may be given.

Indeed, we have measured N-specific antibodies from all the serum samples and the results are shown in Supplementary figure 3. We have highlighted previous PCR-confirmed SARS-CoV-2 infections with black dots and lines, and breakthrough infections with red dots and lines.

Line 155-156: Did the authors analyzed this vaccinee for a possible immunodeficiency?

This is an important concern raised by the Reviewer. However, we did not analyze this participant for possible immunodeficiency since the participant developed high levels of anti-S1 IgG antibodies after the second vaccine dose. Based on the symptom questionnaires the participant suffered from mild SARS-CoV-2 infection and it is possible that for this reason the participant did not develop anti-N IgG antibodies. The lack of or low levels of antibodies after mild symptoms is described e.g. in a study by Petersen et al. (Clin Inf Dis, 2020). In addition, a recent study demonstrated that only 68% of study subjects with breakthrough infection became positive for N-specific antibodies 5 weeks after the infection (Bates et al, 2022, Science Immunology). In addition, based on the original study protocol (ethical permissions) we were not allowed to do any genetic disease related analyses of the vaccinees.

Line 164: ...7.6-9.3 months... Please give the mean or median here.

We have now added a mean time of 8.3 months on line 159.

Line 179: A new paragraph is needed for the second sentence beginning with 'Fifty-nine HCW...'

We have now added a new paragraph as suggested by the Reviewer.

Line 219: Please explain GMT. I could not find the abbreviation explanation also in Figure 5. Please refer to the Figure here and at the end of this paragraph (Line 225).

We have now defined GMT on line 207 in the results, as also mentioned above. We have also added the abbreviation into Figure 5 legend and referred to Figure 5 on line 527.

5. Discussion:

Line 243: containment of the pandemic-> this phrase may be better changed.

We have now rephrased the sentence, line 238-240 "The fast development of efficient COVID-19 vaccines has been an important measure against the pandemic the past year...".

6. Materials and Methods

Please define the N and S1-specific antibody levels.
What does Spike '1' stands for? Why did you add 1 at the end?

We apologize for not defining S1 in the text. We have now modified the immunoassay section: "SARS-CoV-2 spike subunit 1 (S1) and nucleocapsid (N) protein-specific antibodies" on line 309.

Line 321:as described earlier.-> 'before or previously' may be better than earlier.

We agree and have changed earlier to previously.

Line 329: Median Tissue Culture Infectious Dose (TCID50) assay

We have now corrected the text accordingly.

Line 339: Reciprocal of serum dilution?

Reciprocal is the same as multiplicative inverse. For example, a reciprocal of serum dilution 1:360 is 360:1 i.e. 360.

Line 355: UCSF Chimera 1.15 (program....date or version)

We have now added "version 1.15".

Reviewer #3 (Remarks to the Author):

The study "Comparative analysis of BNT162b2, mRNA-1273 and ChAdOx1 COVID-19 vaccine induced antibody responses and the 3rd BNT162b2 vaccine induced neutralizing antibodies against Delta and Omicron variants" by Belik et al is a descriptive study on antibody-responses to Covid-19 vaccination. The authors assess binding- and neutralizing antibody responses to SARS-CoV-2, in vitro, in serum from a cohort of HCWs after covid-19 vaccination.

We thank the Reviewer 3 for the insightful comments and suggestions regarding our manuscript.

Major concern:

It has already been demonstrated that a 3rd dose is beneficial for in vitro neutralizing antibodies against SARS-CoV-2 VoCs, including the Omicron variant (for example Pajon et al NEJM 2022, Garcia-Beltran, Cell 2022, etc). The manuscript by Belik et al verifies these results but do not provide additional in depth information of vaccine-induced immune responses to VoCs.

This statement is valid, however, we want to emphasize that the study by Garcia-Beltran was published online January 6th, 2022, and in print February 3rd, 2022, and the study by Pajon et al. was published online January 26th, 2022, and in print March 17th, 2022, and neither of those were available upon the submission of our manuscript to Nature Communications on December 23, 2021. We would like to point out that there were unfavorable delays in obtaining reviewers for our paper and thus some studies were published while our study was under review.

To understand if the differential dosing intervals matter for epitope specificity after vaccination, it would have been interesting to see a detailed dissection of S-specific B- and T cell responses.

We agree that cell-mediated immune responses are important in protective immunity and immunal memory against SARS-CoV-2 infections. The specific aim of this study was to compare humoral immune responses elicited by different vaccine combinations, and to analyze the duration of neutralizing antibodies in circulation post vaccinations. We hope to be able to look into the cell-mediated immune responses in our upcoming studies.

The authors should acknowledge the discrepancy between protection from infection and protection from severe/fatal covid-19. I.e. that all vaccines appear quite efficient to protect against severe or fatal covid-19 and that this protection is not correlating to antibody levels (whereas protection from infection does).

We thank Review 3 for this noteworthy observation. We have indicated the protective efficacy of vaccinations against severe COVID-19 in Introduction (lines 55-57) and Discussion (lines 262-266), and we have now specified the waning immunity to concern humoral immunity (lines 63-65).